# Quantum-Like Approaches Unveil the Intrinsic Limits of Predictability in Compartmental Models

**DOI:** 10.3390/e26100888

**Published:** 2024-10-21

**Authors:** José Alejandro Rojas-Venegas, Pablo Gallarta-Sáenz, Rafael G. Hurtado, Jesús Gómez-Gardeñes, David Soriano-Paños

**Affiliations:** 1Departamento Administrativo Nacional de Estadística (DANE), Bogotá 111321, Colombia; jarojasv@dane.gov.co; 2Departamento de Física, Facultad de Ciencias, Universidad Nacional de Colombia, Bogotá 111321, Colombia; 3Departamento de Física de la Materia de Condensada, Universidad de Zaragoza, 50009 Zaragoza, Spain; pgallarta@unizar.es; 4GOTHAM Lab, Instituto de Biocomputación y Sistemas Complejos (BIFI), Universidad de Zaragoza, 50018 Zaragoza, Spain; 5Departament d’Enginyería Informática i Matemátiques, Universitat Rovira i Virgili, 43007 Tarragona, Spain

**Keywords:** epidemic dynamics, compartmental models, Doi–Peliti formalism

## Abstract

Obtaining accurate forecasts for the evolution of epidemic outbreaks from deterministic compartmental models represents a major theoretical challenge. Recently, it has been shown that these models typically exhibit trajectory degeneracy, as different sets of epidemiological parameters yield comparable predictions at early stages of the outbreak but disparate future epidemic scenarios. In this study, we use the Doi–Peliti approach and extend the classical deterministic compartmental models to a quantum-like formalism to explore whether the uncertainty of epidemic forecasts is also shaped by the stochastic nature of epidemic processes. This approach allows us to obtain a probabilistic ensemble of trajectories, revealing that epidemic uncertainty is not uniform across time, being maximal around the epidemic peak and vanishing at both early and very late stages of the outbreak. Therefore, our results show that, independently of the models’ complexity, the stochasticity of contagion and recovery processes poses a natural constraint for the uncertainty of epidemic forecasts.

## 1. Introduction

Understanding the temporal dynamics of epidemic outbreaks is critical for pandemic management [1,2]. Classical compartmental models for disease transmission, grounded in the pioneering work of Kermack and McKendrick [3], are considered the keystones of mathematical epidemiology to quantify the public health threat posed by a novel pathogen. Indeed, different indicators, such as the effective reproduction number [4] or the expected outbreak size, have long served as hallmarks for the design of non-pharmaceutical interventions or vaccine rollouts to mitigate the social impact of infectious diseases.

The use of compartmental models for the assessment of epidemic scenarios implicitly assumes that the ultimate consequences of control policies on disease outbreaks can be predicted. However, forecasting the long-term evolution of infectious disease outbreaks still remains a major challenge [5,6], both theoretically [7,8,9] and from a data-driven perspective [10,11].

Focusing on real data, such challenge can be attributed to the unpredictability and intricacies of the variety of biological and social factors neglected in simple compartmental models but ultimately shaping the long-term propagation of infectious diseases [12,13]. Along this line, multimodel forecasting efforts [14,15], integrating predictions from multiple frameworks with different underlying assumptions, have been recently proposed as a solution to partially overcome the former fundamental limitation and to extend the time horizon over which accurate forecasts can be made.

Beyond external factors not included in their formulation, the intrinsic mathematical properties of simple deterministic compartmental models also pose limitations for the reliability of their long-term epidemic forecasts. Recent works have shown that the parameter identification issue [16,17], measuring whether epidemiological parameters can be retrieved when calibrating models with limited data, represents an important source of uncertainty for epidemic forecasts. This issue is exacerbated when noisy points [18] or very early stages of the outbreak [19] are considered for calibration purposes, as trajectory degeneracy ultimately makes divergent predictions compatible with these data. The latter issue hampers the accuracy of the predictions of key quantities in mathematical epidemiology such as the time and size of the epidemic peak or the duration of an epidemic wave [20].

Recognizing the limitations of deterministic models, the role of stochasticity in epidemic models has been also addressed over recent years [21,22,23]. For instance, stochastic Markovian models allow for capturing the uncertain course of epidemic trajectories in small size populations [24,25,26,27] and improving the calibration of epidemic frameworks to real noisy data through the use of adapted Kalman filters [28,29]. In general, accounting for the inherent stochastic dynamics in epidemic processes requires moving from a set of equations governing the time evolution of the expected number of cases to a master equation approach yielding a probabilistic ensemble of epidemic trajectories. One simple approach consists of introducing random fluctuations in the spreading dynamics to improve the compartmental differential equations [30], while more sophisticated frameworks rely on the use of quantum mechanics tools to study different dynamic systems [31,32] or the Doi–Peliti formalism to study the critical behaviour of epidemiological models using the Hamilton–Jacobi equations [33].

Despite these novel approaches, determining the influence of the inherent stochasticity of epidemic processes on forecast uncertainty remains an open problem. To fill this gap, we follow previous works and propose a quantum-like formalism to model epidemic dynamics by extending the Doi–Peliti approach [34,35] to the classical susceptible–infected–susceptible (SIS) and susceptible–infected–recovered (SIR) models. By leveraging the Doi–Peliti formalism, our paper aims at (i) unravelling hidden behaviours in classical deterministic compartmental models and (ii) showcasing how stochasticity shapes the uncertainty of epidemic outbreaks. In particular, the main contribution of this study is to reveal that the stochastic nature of epidemic processes hinders obtaining faithful long-term forecasts on the magnitude and position of the epidemic peak at early stages of an epidemic outbreak.

This article is organized as follows. We first introduce the theoretical formalism of our work in Section 2, including the basic rules to describe the system and the Master Equation. Then, in Section 3, we present the main results of our work, related to the simulation of the Master Equation, the probability of finding minor outbreaks and the stochastic determinants for the uncertainty of epidemic forecasts. Finally, we discuss the implications of our findings and future research venues in Section 4.

## 2. Doi–Peliti Approach to Compartmental Models

In this section, we present the theoretical background of our work, both the classical compartmental models and Doi–Peliti formalism, with equations that lead to the description of this new approach. In both cases, we consider a closed population with *N* individuals, thus neglecting any changes in the population size as a result of birth–death processes. Regarding the structure of contacts, we restrict to the simplest scenario and follow a mean-field assumption considering well-mixed populations.

### 2.1. Deterministic Equations for the SIS and SIR Models

The most usual way to tackle the modelling of SIS and SIR dynamics is to consider a set of ordinary differential equations (ODEs) governing the time evolution of the expected occupation of the different compartments. In the simplest case, the SIS model assumes that each individual can be either in the Susceptible or in the Infected state and that transitions between them correspond to contagion and recovery processes. In particular, Susceptible individuals become infectious at a β rate upon contact with Infected individuals. Moreover, Infected individuals recover at a γ rate, without acquiring any immunity against the circulating pathogen. Denoting the occupation of the Susceptible (Infected) compartment by *S* (*I*), the deterministic equations capturing the SIS dynamics are as follows:(1)$$\frac{d}{dt}S=-\beta S\frac{I}{N}+\gamma I,\;\frac{d}{dt}I=\beta S\frac{I}{N}-\gamma I.$$

The SIR model, instead, accounts for those infections conferring immunity to the host upon recovery. This aspect is included by assuming that infectious individuals recover at a γ rate and enter into a new compartment, the Removed state *R*, rather than returning to the Susceptible state. From these simple rules, the differential equations of the SIR model read as follows:(2)$$\frac{d}{dt}S=-\beta S\frac{I}{N},\;\frac{d}{dt}I=\beta S\frac{I}{N}-\gamma I,\;\frac{d}{dt}R=\gamma I.$$

### 2.2. The Doi–Peliti Master Equation

Going beyond the traditional deterministic approach, epidemic dynamics can be interpreted as stochastic birth–death processes. In these models, individuals transition (or ‘die’) from one compartment to ‘be born’ into another. In this context, Doi–Peliti formalism [34,35] takes advantage of the quantum field theory to build a Markovian Master Equation (MME). This approach requires two fundamental components: the vectors, |φ〉, describing the dynamical state of the system, and the creation–annihilation operators, *a*, $a^{†}$, which create or annihilate individuals, respectively, in the compartments described in the epidemic models.

Regarding the first component, we follow a probabilistic approach considering that the state of our system |φ〉 exists in the space in which the elements of the basis |ϕ〉 span, with each one representing a possible configuration of the model under consideration. Mathematically, we assume the following:(3)$$|\varphi \rangle =\sum_{\phi }P\left(\phi \right)|\phi \rangle \;.$$

The elements |ϕ〉 of the basis depend on the compartmental model and are explained below for both the SIS and the SIR models. The second component concerns the ladder operators for each compartment, *a* and $a^{†}$, creating or removing individuals, respectively. Assuming that |x〉 represents the element of the basis corresponding to an occupation number *x* of a given compartment, the previous operators are defined as follows:(4)$$a^{†}|x\rangle =|x+1\rangle ,$$
(5)a|x〉=x|x−1〉.Likewise, these operators follow some quantum mechanics principles: their eigenvalues allow us to write the states as $|x\rangle ={\left(a^{†}\right)}^{x}|0\rangle$, and they satisfy the usual commutation rule $\left[a,\;a^{†}\right]=aa^{†}-a^{†}a=\hat{I}$. Furthermore, it is also possible to define a number operator, $\hat{n}=a^{†}a$, which returns the occupation number of the state $\hat{n}|x\rangle =x|x\rangle$.

With these two ingredients, one can construct the Hamiltonian-governing different dynamics in systems with many-body interactions as outlined in [33,36]. Regardless of the chosen dynamics, the Doi–Peliti approach allows us to capture the evolution of the system state, |φ〉, using a backward master equation (BME) [34,35], analogous to Schrödinger’s equation with an imaginary time as follows:(6)$$\frac{d}{dt}|\varphi \rangle =H|\varphi \rangle .$$

### 2.3. The Doi–Peliti Approach to the SIS Model

In an SIS dynamics with a closed population of *N* individuals, the number of infected individuals *I* provides enough information to describe the state of the system $|\varphi_{SIS}\rangle$, given the constraint S=N−I. Thus, the state $|\varphi_{SIS}\rangle$ can be written as a combination of all possible occupation numbers, ${\left\{|I\rangle \right\}}_{I=0,\ldots ,N}$, forming the following basis of the system:(7)$$|\varphi_{SIS}\rangle =\sum_{I}P\left(I\right)|I\rangle .$$In the former linear combination, the coefficients $P\left(I\right)=\langle I|\varphi_{SIS}\rangle$ measure the probability associated with each occupation number, unequivocally defining the state $|\varphi_{SIS}\rangle$ by the set ${\left\{P(I)\right\}}_{I=0,\ldots ,N}$.

To construct the Hamiltonian governing SIS dynamics, we consider different ladder operators acting on each compartment. We consider that the operators *a*, $a^{†}$ act on the susceptible states, whereas *b*, $b^{†}$ act on the infectious ones. Therefore, the operator $\left(ab^{†}\right)$ models contagion processes, creating an infectious individual and removing a susceptible one, whereas $\left(ba^{†}\right)$ captures recovery processes. Consequently, the Hamiltonian of the SIS dynamics, $H_{\mathrm{SIS}}$, reads as follows:(8)$$H_{\mathrm{SIS}}=-\frac{\beta }{N}n_{I}\left(n_{S}-ab^{†}\right)-\gamma \left(n_{I}-ba^{†}\right).$$

### 2.4. The Doi–Peliti Approach to the SIR Model

The SIR model requires a basis that accounts for the occupation numbers of two of the three compartments (*S*, *I*, and *R*). Here, without loss of generality, we take the infected and susceptible occupation numbers to fully capture the system composition. Therefore, we define the state basis as ${\left\{|S,\;I\rangle \right\}}_{S,\;I=0,\ldots ,N;\;S+I\leq N}$. Hence, the state $|\varphi_{SIR}\rangle$ is expressed as follows:(9)$$|\varphi_{SIR}\rangle =\sum_{S,\;I}P\left(S,I\right)|S,\;I\rangle ,$$
with $P\left(S,I\right)=\langle S,\;I|\varphi_{SIR}\rangle$. Analogous to the SIS case, we assume that the set $\left\{P(S,I)\right\}$ defines $|\varphi_{SIR}\rangle$, simplifying the notation and the representation of the states.

To construct the SIR Hamiltonian, we must define the ladder operators *c* and $c^{†}$ acting on the recovered compartment. In the SIR model, the transition of an infected individual moving to the recovered state $\left(bc^{†}\right)$ replaces the transition to the susceptible state of the SIS model. Then, the Hamiltonian $H_{\mathrm{SIR}}$ is as follows:(10)$$H_{\mathrm{SIR}}=-\frac{\beta }{N}n_{I}\left(n_{S}-ab^{†}\right)-\gamma \left(n_{I}-bc^{†}\right).$$

## 3. Results

In this section, we present the main results of our work, derived from simulations of the Master Equation discussed earlier. The results are organized into three subsections: the dynamics of both models (Section 3.1), a formal analysis and computational solution for the probability of no-outbreak results (Section 3.2), and an examination of the predictability problem using entropy measures of temporal dynamics (Section 3.3).

### 3.1. Dynamics of the Doi–Peliti Master Equation

Figure 1 summarizes the two different approaches of classical compartmental models, highlighting the conceptual differences between both the determistic ODEs for the SIS and SIR models and their corresponding equations based on the Doi–Peliti approach.

To obtain the time evolution of epidemic outbreaks in both the SIS and the SIR model under the Doi–Peliti approach, we should use the matrix representation of the Hamiltonian operator and compute its elements $H_{x,\;x^{\prime }}=\langle x|H|x^{\prime }\rangle$ capturing the transitions between the possible configurations |x〉 and $|x^{\prime }\rangle$ for each model. Once the Hamiltonian is defined, the analytical solution of Equation (Equation 6) can be readily obtained as follows [37]:(11)$$|\varphi \left(t\right)\rangle =e^{Ht}|\varphi \left(0\right)\rangle ,$$
where expHt represents the propagator of each dynamics.

The computation of the system propagator for each time *t* might be cumbersome, especially for high-dimensional systems where the diagonalization of the Hamiltonian is computationally expensive. To overcome this limitation, we rely on the Markovian property of the master equation and consider the time evolution of the system over multiple discrete time steps of duration Δt. In each time step, the state of the system is updated as follows:(12)$$|\varphi \left(t+\Delta t\right)\rangle =e^{H\Delta t}|\varphi \left(t\right)\rangle .$$Therefore, the time evolution of the system can be obtained as the subsequent action of a single propagator $e^{H\Delta t}$ on the updated state according to Equation (Equation 12), thus saving the computational time associated with the computation of the propagator. Throughout the manuscript, we assume Δt=0.1.

As explained above, the evolution of epidemic outbreaks in the SIS model is fully characterized by monitoring the time evolution of the probabilities P(I(t)) of finding *I* individuals in the infected compartment at time *t*. Without any loss of information, let us instead focus on the probability of finding a fraction of population $\rho_{I}\left(t\right)$ in such a compartment at time *t*, with $\rho_{I}\left(t\right)=I\left(t\right)/N$. Figure 2a represents the evolution of this set of probabilities for an epidemic triggered by a single infectious individual in a population of $N_{SIS}=1000$ individuals, characterized by β=0.6, γ=0.1, thus with a basic reproduction number $R_{0}=\beta /\gamma =6$. We also represent the analytical solution of the deterministic ODE, Equation (Equation 1) in the same figure, governing the evolution of the SIS model (dashed line of Figure 2a).

The comparison between both probabilistic and deterministic approaches reveals the wealth of information typically overlooked by classical deterministic models. First, epidemic uncertainty is not uniform across time, being maximal at intermediate stages. For instance, at t=15, the deterministic equations predicts a widespread epidemic ($\rho_{I}≃0.4$), whereas the probabilistic ensemble of trajectories also shows a significant probability of finding a small epidemic outbreak $\rho_{I}≃0$. The latter shows how the classical indicator, i.e., the expected fraction of population in the infected state, might not be a representative quantity to capture the stochastic transient dynamics of epidemic outbreaks. This behaviour cannot be reproduced by the deterministic model. Remarkably, this uncertainty shrinks around the deterministic value at later stages t>30, showing the robustness of classical deterministic approaches in determining the metastable epidemic state of the system. Note, however, that the steady state of the stochastic system is always $\rho_{I}^{\infty }=0$. This occurs because the absence of infected individuals represents an absorbing state with a probability of occupation always increasing over time as a result of stochastic fluctuations destabilizing the metastable epidemic state.

For the SIR model, we characterize the evolution of epidemic outbreaks by monitoring the occupation of both the infected and the recovered compartments. As the state of the system is described by $\left\{P(S,I)\right\}$ (see Section 2.4), we should compute the marginal probabilities $\left\{P(I)(t)\right\}$ and $\left\{P(R)(t)\right\}$ as follows:(13)$$\begin{array}{ccc} P(I) & = & \sum_{S}P\left(S,\;I\right), \end{array}$$(14)$$\begin{array}{ccc} P(R) & = & \sum_{S,I|S+I=N-R}P\left(S,\;I\right). \end{array}$$

Figure 2b,c represent the time evolution of the infected and recovered compartments, respectively, considering a population of $N_{SIR}=100$ individuals due to computational memory limits, including the deterministic solution of Equation (Equation 2) with dashed lines. In these plots, we can observe that the Doi–Peliti approach reproduces the wave-like behaviour of the epidemic outbreaks under the SIR model. As in the case of the SIS model, we observe that epidemic uncertainty is not uniform across time and that the deterministic trajectories capture the time evolution of the expected value of the probability distributions yielded by the Doi–Peliti approach. Additionally, we observe a bimodal probability density function of the recovered individuals. While the region closer to the deterministic equations captures major epidemic outbreaks, the probabilistic cloud with negligible recovered population is a consequence of the stochastic effects driving the system to the absorbing state before any epidemic is observed in the population.

Our results reveal that epidemic uncertainty is not uniform in a single epidemic trajectory for both the SIS and SIR models. To fully characterize the impact of stochasticity on epidemic dynamics, we now analyze the uncertainty of the order parameters of both models as a function of the basic reproduction number $R_{0}$ of the disease, as shown in Figure 3. For the SIS model, Figure 3a represents the prevalence of the disease in the metastable epidemic state, at t=1000 days, with a deterministic value that is given by $\rho_{I}^{SIS,met}=1-R_{0}^{-1}$. Conversely, for the SIR model, we analyze in Figure 3b the attack rate of the disease $\rho_{R}^{\infty }$ defined as the fraction of recovered individuals at equilibrium. The deterministic value for this parameter is obtained by solving the implicit equation $\rho_{R}^{\infty }=1-s_{0}e^{-R_{0}\left(\rho_{R}^{\infty }-r_{0}\right)}$, where $s_{0}$ and $r_{0}$ represent the initial proportion of susceptible and recovered individuals in the population. In both cases, the Doi–Peliti framework proves that there is a high probability of observing the order parameters predicted by the deterministic ODEs. Notably, as previously stated, the SIR model reveals a notable feature: a non-negligible probability of having a minor outbreak in the stationary state, even for values of $R_{0}>1.0$, as shown in Figure 3b. We further explore this phenomenon in Section 3.2.

Qualitatively, Figure 3 reveals that the uncertainty of the predictions for the order parameter is not uniform but instead varies as a function of the basic reproduction number of the disease $R_{0}$. To quantify such behaviour, we compute the entropy *H* of the marginal probability distributions for the occupation of a single compartment *X* in each epidemic model, denoted by P(X) as follows:(15)$$H=-\sum_{X}P\left(X\right)\log P(X),$$
where *X* stands for the infected (recovered) compartment *I* (*R*) in the case of the SIS (SIR) model.

Figure 3c represents the evolution of entropy for both models as a function of $R_{0}$ and the number of individuals initially infected $I_{0}$. First, we observed that entropy does not follow a monotonic dependence on $R_{0}$ for the SIR model. Instead, entropy reaches a maximum value and then drops as $R_{0}$ increases. This behaviour is driven by the presence of a no-outbreak probability, which significantly influences the system when $R_{0}∼1.0$, leading to a greater uncertainty. As $R_{0}$ grows further, the probability of a minor outbreak diminishes, and the cloud probability around $\rho_{R}^{\infty }∼0$ shrinks. Analogously, for the SIS model, there is a high probability of falling into the absorbing state in the vicinity of $R_{0}≃1$, which starts decreasing with $R_{0}$, favoring the occupation of the epidemic metastable state to the detriment of the absorbing state. This phenomenon is reflected by the increase in entropy observed for the SIS model with $R_{0}$, which might give rise to a non-monotonic behaviour when the occupation of the metastable epidemic state becomes dominant over the absorbing one. Note that the entropy for the SIS model is analyzed at t=1000 as in the steady state $H_{SIS}^{\infty }=0$, provided the absorbing state is the single equilibrium of the dynamics.

### 3.2. The Probability of No-Outbreak in the Doi–Peliti Formalism

The interplay between the stochastic nature of compartmental models and the existence of absorbing states has been long studied in the mathematical epidemiology field. In 1955, Whittle [38] derived the probability to find a minor outbreak caused by a pathogen with a given reproduction number $R_{0}$. Using the theory of branching processes, the probability of extinction of an outbreak given a number of $I_{0}$ individuals, $\pi_{NO}\left(I_{0}\right)$, is fulfilled as follows:(16)$$\pi_{NO}\left(I_{0}\right)=\frac{\beta S_{0}I_{0}/N}{\beta S_{0}I_{0}/N+\gamma I_{0}}\pi_{NO}\left(I_{0}+1\right)+\frac{\gamma I_{0}}{\beta S_{0}I_{0}/N+\gamma I_{0}}\pi_{NO}\left(I_{0}-1\right)\;.$$Note that the two terms in the previous equation can be related to $H_{SIR}$ in the Doi–Peliti approach. In particular, $\beta S_{0}I_{0}/N$ corresponds to the rate of transition $H_{|S_{0},I_{0}\rangle ,|S_{0}-1,I_{0}+1\rangle }$, whereas the term $\gamma I_{0}$ corresponds to $H_{|S_{0},I_{0}\rangle ,|S_{0},I_{0}-1\rangle }$.

Considering a small number of infectious individuals, we can assume $S_{0}≃N$, turning Equation (Equation 16) into the following:(17)$$\pi_{NO}\left(I_{0}\right)=\frac{R_{0}}{R_{0}+1}\pi_{NO}\left(I_{0}+1\right)+\frac{1}{R_{0}+1}\pi_{NO}\left(I_{0}-1\right)$$For the simplest case $I_{0}=1$, and bearing in mind that $\pi_{NO}\left(0\right)=1$, since no outbreak can take place without the initial infectious individuals, the equation is reduced to a quadratic form with the following roots:(18)$$\frac{R_{0}}{R_{0}+1}\pi_{NO}{\left(1\right)}^{2}-\pi_{NO}\left(1\right)+\frac{1}{R_{0}+1}=0⟶\pi_{NO}\left(1\right)=\left\{\;\begin{array}{ccc} \; & \;\;\;1 & R_{0}<1\\ \; & \frac{1}{R_{0}} & R_{0}>1 \end{array}\right.$$

Finally, introducing a tree-like assumption [22] in the presence of multiple initially infected individuals, i.e., $\pi_{2}={\left(\pi_{1}\right)}^{2}$, we can generalize the former equation as follows:(19)$$\pi_{NO}\left(I_{0}\right)=\left\{\;\begin{array}{ccc} \; & \;\;\;\;\;\;\;\;1 & R_{0}<1\\ \; & {\left(\frac{1}{R_{0}}\right)}^{I_{0}} & R_{0}>1 \end{array}\right.$$

To validate this theoretical expression, we perform agent-based simulations relying on the τ-leap algorithm. In particular, we consider a population of $N=10^{5}$ individuals and compute the probability of not observing an outbreak by varying the initial number of infected individuals $I_{0}$ and the basic reproduction number of the pathogen $R_{0}$. The probability of no outbreak $\pi_{NO}\left(I_{0}\right)$ is computed as the fraction of simulations giving rise to minor outbreaks with little impact on the population. To compute such quantity, we classify an epidemic trajectory as a minor outbreak when less than 20% of the expected attack rate of the major outbreak, $R_{\infty }^{det}$, has been infected throughout the dynamics. Mathematically, we assume that the attack rate $R_{\infty }^{minor}$ of minor outbreaks fulfils $R_{\infty }^{minor}\leq 0.2R_{\infty }^{det}+I_{0}$. Since no major outbreak can be reached when $R_{0}<1$, we assume $R_{\infty }^{minor}\leq 0.2N+I_{0}$ in this case. Figure 4 shows that the theoretical expression fairly captures the results from the stochastic simulations. Note that the choice of $R_{minor}^{\infty }$ is somehow arbitrary as there is not a unique way of defining the attack rate corresponding to a minor outbreak in the SIR model as a function of $R_{0}$. Nonetheless, we have checked that the results remain consistent under other choices for $R_{minor}^{\infty }$.

Despite the former agreement, performing agent-based simulations comes with a high computational cost. Obtaining significant results requires considering very large populations to avoid finite size effects, and many outbreaks should be simulated to attain enough statistics to compare them with the theoretical predictions. In contrast, the Doi–Peliti equations can be readily leveraged to compute the probability of minor outbreaks. To do so, we must restrict ourselves to the region of low attack rates and integrate the probability density function of the recovered individuals once the steady state has been reached, fixing the upper bound of integration to the values of $R_{\infty }^{minor}$ considered in the agent-based simulations. Figure 4 confirms that the Doi–Peliti equations allow for characterizing the probability of observing minor outbreaks in the population without the need to perform any agent-based simulations.

### 3.3. The Predictability Problem of the SIR Model

Apart from capturing the probability of minor outbreaks, the Doi–Peliti equations can be leveraged to quantify how the inherent stochasticity of epidemic processes shapes the uncertainty of forecasts during an epidemic outbreak. This uncertainty does not come from the model complexity or trajectory degeneracy in the space of parameters but instead is a consequence of the existence of an underlying probabilistic ensemble of trajectories which can be generated with fixed epidemiological parameters and initial conditions.

To tackle this problem, we generate a synthetic trajectory with the deterministic equations of the SIR model and assume that these data represent the actual time evolution of an epidemic outbreak with $R_{0}=6.0$. To address how epidemic uncertainty changes over time, we run Equation (Equation 12), assuming that the initial conditions correspond to the epidemic state of the system across different time points of the epidemic trajectory. Note that, for each time step, starting the epidemic outbreak from an epidemic point resembles the effect of measurements in quantum mechanics (epidemic data), which change the probabilistic state of the system to a well-defined one.

Figure 5 represents the time evolution of the probability density functions for the density of infected individuals assuming four different initial times: $t_{0}=0$ (Panel a), $t_{0}=t_{peak}/2$ (Panel b), $t_{0}=t_{peak}$ (Panel c), and $t_{0}=20$ (Panel d). Several key insights can be drawn from this figure. First, the width of the probability cloud decreases as the initial condition time increases, indicating a reduction in the uncertainty present in the system. Moreover, the Doi–Peliti formalism reveals vastly different epidemic impacts around the epidemic peak, which shows how deterministic approaches overlook many possible epidemic scenarios [39]. Additionally, in Panels b–d, the probability of no outbreak disappears due to the large initial conditions.

To further quantify the time evolution of the predictability of the outbreak, we consider different initial time points and compute the entropy of the generated probability distributions of infected individuals at the epidemic peak, $H_{inf}\left(t_{peak}\right)$. The latter reads as follows:(20)$$H_{inf}\left(t_{peak}\right)=-\sum_{I}P\left(I(t_{peak})\right)\log P\left(I(t_{peak})\right)$$

For the sake of generality, we perform the former analysis by considering several epidemic outbreaks characterized by different $R_{0}$ values. For a fair comparison between epidemic scenarios, we consider the time points $\tilde{t_{0}}$ in terms of the relative difference between the time where forecasts are made and the time of the epidemic peak. In particular, we define $\Delta {\tilde{t}}_{0}=\left(t_{0}-t_{peak}\right)/t_{peak}$ with $\Delta {\tilde{t}}_{0}\in \left[-1,\;0\right]$.

Figure 6a represents the time evolution of the epidemic uncertainty of the peak $H_{inf}\left(t_{peak}\right)$ as a function of the time taken for forecasting purposes $\Delta {\tilde{t}}_{0}$. There, we observe that the entropy $H_{inf}$ initially rises around $\Delta \tilde{t}\approx -1$, reaches a maximum, and then drops to zero at $\Delta \tilde{t}=0$. The initial rise in entropy may seem counterintuitive at first sight, but it is linked to the no-outbreak probability. When $I_{0}$ is small ($\Delta {\tilde{t}}_{0}\approx -1$), there is a non-negligible probability of no outbreak ($P_{0}\equiv \pi_{NO}\left(I_{0}\right)>0$) present throughout the time series (Figure 5a). As $\Delta {\tilde{t}}_{0}$ grows, $P_{0}$ decreases, leading to a less defined state (inset of Figure 6a). This loss of information about the state is reflected in the slight rise in the entropy. Once the initial conditions discard minor outbreaks, the uncertainty of forecasts decreases as they are made closer to the epidemic peak, thus replicating the observed phenomena in classical deterministic models [20].

In Figure 6b, we represent
(21)$$\Delta {\tilde{t}}_{H_{inf}^{max}}=\frac{\left(t_{H_{inf}^{max}}-t_{peak}\right)}{t_{peak}}$$
against $\Delta {\tilde{t}}_{0}$, measuring the relative position of the maximum entropy found in the epidemic trajectory, $t_{H_{inf}^{max}}$, compared to the time at which the peak of infected individuals occurs. $\Delta {\tilde{t}}_{H_{inf}^{max}}<0$ indicates that the maximum of entropy occurs before the peak of contagions, and $\Delta {\tilde{t}}_{H_{inf}^{max}}>0$ implies that the uncertainty is higher at later stages. At early times of the outbreak, we find $\Delta {\tilde{t}}_{H_{inf}^{max}}<0$, indicating that maximum uncertainty precedes the infection peak. Regardless of the basic reproduction number $R_{0}$, the latter position is delayed as forecasts are performed at later stages. Interestingly, such delay is not linear with the forecasting time, highlighting a complex interplay between the underlying stochastic dynamics and the probabilistic output of epidemic processes in determining the uncertainty associated with epidemic forecasts.

## 4. Discussion

Compartmental models are widely used to characterize mathematically epidemic outbreaks, obtain short-term forecasts on their evolution, and design control policies to mitigate their impact on society [40,41]. These models usually rely on deterministic approaches based on ODEs to produce the expected time evolution of the number of cases in the population. Therefore, most of the epidemic trajectories obtained do not account for the stochastic nature of the underlying epidemiological processes driving the onset of infectious diseases. To overcome this limitation, in this work, we have developed a quantum-like approach, based on a Hamiltonian formulation of both the SIS and SIR dynamics through Doi–Peliti equations. Our approach provides a probabilistic description of the ensemble of possible epidemic trajectories yielded by the stochasticity of both contagion and recovery processes.

The analysis of the probabilistic cloud of infections reveals interesting phenomena which cannot be observed through the lens of deterministic models. For the SIR model, we have first shown how such clouds typically present two disjoint areas with high density of trajectories, corresponding to the propagation of major and minor outbreaks in the population. Indeed, our results showed that the Doi–Peliti equations yield a fair estimation of probability that a given pathogen generates a minor outbreak without the need of performing computationally expensive agent-based simulations.

Focusing on major outbreaks, our results show that the uncertainty of epidemic trajectories is not uniform across time, being maximal around the peak of contagions. This finding poses theoretical constraints to the accuracy of long-term forecasts on the position and magnitude of the epidemic peak. Indeed, for several pathogens with different infectiousness, we show how the forecasts uncertainty around the epidemic peak is only reduced when those are made considerably close to its position. Therefore, our results prove that the reliability of epidemic forecasts is not only limited by the intrinsic complexity of compartmental models [42,43] but also by the stochasticity of the epidemiological processes determining the onset of pathogens in the population.

Regarding the limitations of the present work, one significant challenge is the amount of memory needed to store the propagators in the equations. These propagators are typically sparse as transitions between many epidemic states are not allowed given the definition of the ladder operators. Nonetheless, memory demands can become a limiting factor to analyze epidemic outbreaks in large size populations as the size of the propagators scales with the number of individuals *N*. From a practical point of view, both the compartmental model describing the course of the disease and the assumptions on the structure of contacts should be improved to use our formalism in a real epidemic scenario [44]. Factors such as political interventions, mobility patterns, demographic information, or spatial structure must be incorporated to account for more complex behaviours within the model [45]. However, yet improving the realism of the model, such refinements would also enlarge the propagators of the system considerably, as more information would be required to capture the evolution of any possible microstate in the system.

In summary, our study highlights that the mathematical characterization of epidemic dynamics through deterministic ODEs misses very rich phenomena arising from the stochasticity of epidemic outbreaks. We believe our theoretical framework provides a solid ground for the future development of more complex models, leveraging advanced probabilistic models to refine our understanding of epidemic phenomena reported in agent-based simulations. For instance, the extension of this framework to networked populations could improve our understanding of the Griffiths phases [46,47] appearing close to the epidemic threshold in complex networks. Likewise, the Doi–Peliti equations on metapopulations could serve as a benchmark to characterize the so-called invasion threshold [48,49] of pathogens driven by epidemic mobility without the need for agent-based simulations. This ongoing development is promising for enhancing the predictive capabilities and improving our responses to future outbreaks, contributing to better public health outcomes and more accurate interventions.

## Figures and Tables

**Figure 1 entropy-26-00888-f001:**
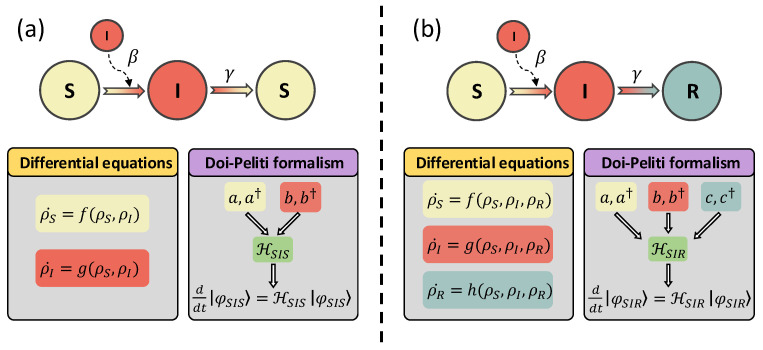
Comparison between theoretical approaches to compartmental models based on differential equations and the Doi–Peliti equations for the SIS (Panel (**a**)) and SIR (Panel (**b**)) models. The classical approach consists of deriving a set of deterministic ODEs governing the time evolution of the expected occupation of each compartment *m*, denoted by $\rho_{m}$. Conversely, the Doi–Peliti approach involves a quantum-like approach, constructing the Hamiltonian for both dynamics from the ladder operators determining the occupation of each compartment and using the time-dependent Schrödinger equation for the evolution of the dynamical state of the system.

**Figure 2 entropy-26-00888-f002:**
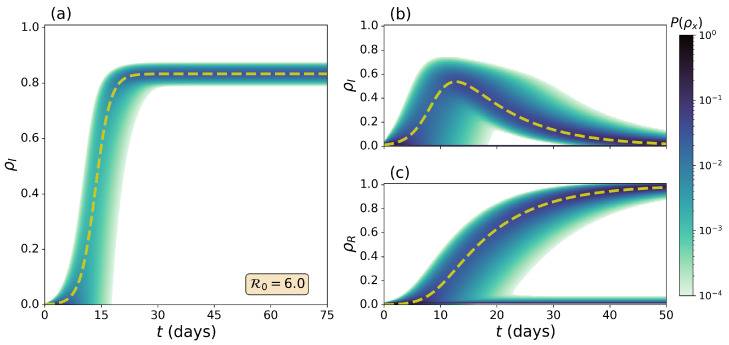
(**a**): Time evolution of the probability of finding a fraction $\rho_{I}$ of individuals in the infected state $P(\rho_{I})$ (color code) for an SIS dynamics. (**b**,**c**) The time evolution of the probability of finding a fraction $\rho_{m}$ of the population in the compartment *m*, $P(\rho_{m})$ (color code) for an SIR dynamics. The compartments shown are (**b**) the Infected compartment and (**c**) the Recovered compartment. The deterministic solutions of Equations (Equation 1) and (Equation 2) (dashed lines) are shown over the cloud probability to compare both frameworks. In panel (**a**), we consider a population of $N_{SIS}=1000$ individuals and, in panels (**b**,**c**), a population of $N_{SIR}=100$ individuals. In all panels, we fix the contagion rate to β=0.6 and the recovery rate to γ=0.1, yielding a basic reproduction number $R_{0}=6$.

**Figure 3 entropy-26-00888-f003:**
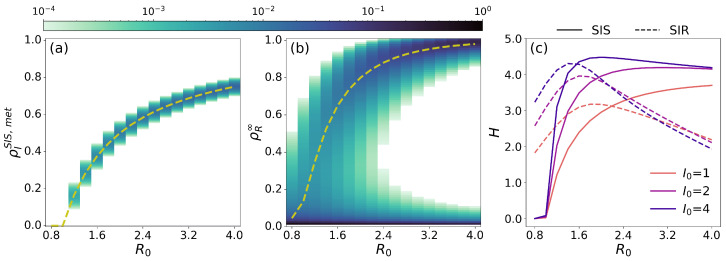
(**a**,**b**): Evolution of the cloud probability of the order parameters as $R_{0}$ evolves, using the prevalence of the metastable epidemic state computed at t=1000 days $\rho_{I}^{SIS,met}$ in the SIS model (Panel (**a**)) and the attack rate $\rho_{R}^{\infty }$ in the SIR model (Panel (**b**)). The deterministic solution for the order parameters of both models is also shown in both panels with dashed lines. (**c**) The evolution of the entropy as a function of $R_{0}$ for the SIS (solid lines) and SIR (dashed lines) and three different initial conditions, varying the number of initially infected individuals $I_{0}$ (color code). For the SIS model, we consider a population of $N_{SIS}=1000$ individuals and, for the SIR, a population of $N_{SIR}=100$ individuals. In all panels, we fix the recovery rate to γ=0.1 and modify the infection rate β, such that $R_{0}\in \left[0.8,4.0\right]$.

**Figure 4 entropy-26-00888-f004:**
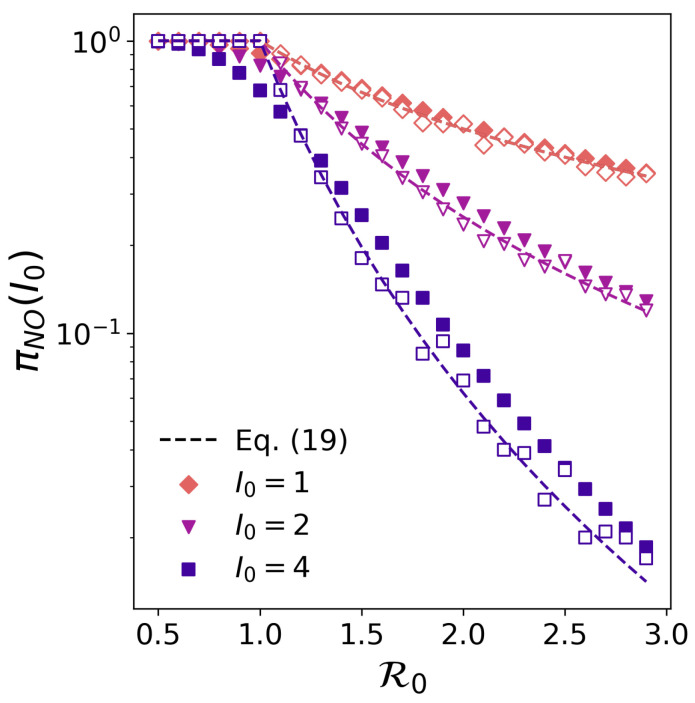
Probability of generating minor epidemic outbreaks as a function of the basic reproduction number $R_{0}$ and the number of individuals initially infected $I_{0}$ (symbols and color code). Dashed lines represent theoretical estimations obtained from Equation (Equation 19). Filled dots represent integration in the region of low attack rates (see text for details) of the probability density function for recovered individuals obtained through the Doi–Peliti equations. Empty dots represent the results from agent-based simulations using the τ-leap algorithm. We simulate n=1000 epidemic trajectories, considering that a minor outbreak is characterized by an attack rate $R_{\infty }^{minor}<0.20N+I_{0}$ if $R_{0}<1$ and $R_{\infty }^{minor}<0.05R_{\infty }+I_{0}$ if $R_{0}>1$.

**Figure 5 entropy-26-00888-f005:**
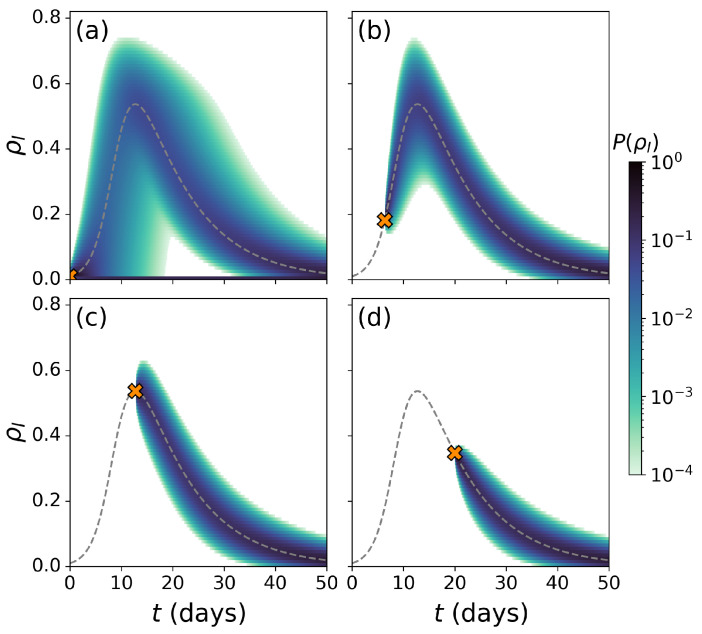
Time evolution of the probability of finding a fraction $\rho_{I}$ of individuals in the infected compartment *I*, $P(\rho_{I})$ (color code), for the SIR dynamics of a pathogen with $R_{0}=6$ propagating across a population of N=100 individuals. The initial conditions to run the Doi–Peliti equations (cross symbol) are set according to the values of the deterministic epidemic trajectories at different stages $t_{0}$ of the outbreak: (**a**) $t_{0}=0$, (**b**) $t_{0}=t_{peak}/2$, (**c**) $t_{0}=t_{peak}$ and (**d**) $t_{0}=20$.

**Figure 6 entropy-26-00888-f006:**
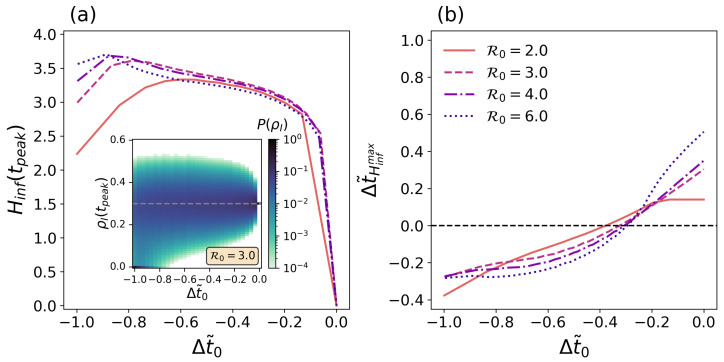
(**a**): Entropy of the marginal distribution of infected individuals at the epidemic peak $H_{inf}\left(t_{peak}\right)$ as a function of the time from which forecasts are made $\Delta {\tilde{t}}_{0}$ and the reproduction number of the pathogen $R_{0}$ (color code). The inset panel shows the time evolution of the marginal probability distribution for the fraction of population in the infected state at the epidemic peak $P(\rho_{I}\left(t_{peak}\right))$ (color code). (**b**) Relative position of the highest entropy observed in the epidemic trajectory $\Delta {\tilde{t}}_{H_{inf}^{max}}$ as a function of $\Delta \tilde{t}$ and the reproduction number of the pathogen $R_{0}$ (color code). In all panels, time is measured in relative units to the position of the epidemic peak (see text for further details).

## Data Availability

The original contributions presented in the study are included in the article, further inquiries can be directed to the corresponding author/s.

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
