# Peer review of "Quantum-Like Approaches Unveil the Intrinsic Limits of Predictability in Compartmental Models"

_entropy, 2024, doi:10.3390/e26100888_

Round 1
Reviewer 1 Report
Comments and Suggestions for Authors
Thank you for the opportunity to review this manuscript and congratulations on this work. I have two brief comments.
1. I would like to see incorporation of real-world pandemic data to corroborate this study (either in parameter estimates or direct comparison). Simulated data is only so convincing.
2. The following paper is closely aligned with the conclusions of this manuscipt. Please cite / incorporate as needed. https://www.frontiersin.org/journals/epidemiology/articles/10.3389/fepid.2024.1389617/full
Reviewer 2 Report
Comments and Suggestions for Authors
Review on the paper “entropy-3213922”
The article proposes a solution beyond the widespread deterministic susceptible-infected-susceptible (SIS) and susceptible-infected-recovered (SIR) models. It is clear that epidemic processes can be formulated much more in probability. It is ingenious to transfer the Doi-Peliti equation to a Markovian Master equation, including the stochastic behavior. It is promising to introduce the creation and destruction operators into the description. This idea, derived from quantum theory, the second-quantal formalism, is worth further discussion. I cannot verify the numerical calculations, but the presented results are believable and probably good.
What I miss is that at least the directions for further development of the model should be mentioned in the Discussion. What happens if the number of the considered population changes with emigration and immigration, if there is natural death? What happens when population groups are far from each other, isolated to some degree? They are missing from both the deterministic and the present model. I don't think this extension of the problem needs to be solved here, but it would still be nice to show a distant target.
The article is well-written, understandable, and interesting. I recommend its acceptance for publication.
Reviewer 3 Report
Comments and Suggestions for Authors
In this work, the authors extend the classical deterministic SIS and SIR models to stochastic models, employing an elegant quantum-like formalism known as the Doi-Peliti approach, which utilizes ladder operators to model the probabilistic dynamics of different compartments. They further show that the uncertainty in epidemic trajectories is not uniform over time, reaching its maximum around the peak of contagions. This observation offers a plausible explanation for the increased forecast uncertainty typically observed near the epidemic peak, which is responsible for the key limitation in practical forecasting tasks.
The paper is very well written with informative figures and well-organized text. I am fully convinced that the work presents intriguing physics that is not only appealing to the academic community but also carries significant practical implications. In fact, this study may open up several promising directions worth exploring. For example, could there be new emerging strong-disorder phenomena (akin to the Griffiths phase)? Is it possible to design strategies that not only reduce the outbreak size but also improve predictability (as the latter is also important for public health)? Given the strengths of the work, I strongly recommend its publication, with no major revisions needed (except for the following minor typos):
1. Caption of Fig. 1: SIR (models) -> SIR (Panel b)
2. While the authors mention “…We also represent in the same figure the analytical solution of the deterministic ODE…”, I do not see any caption description of which is the analytical solution (the dashed lines?) in figure 2.
3. Line 188: the text shows “(see Materials and Methods)”, but I could not find a Materials and Methods section.
4. Line 328: Griffits phases-> Griffiths phases
Additionally, I suggest that the paper be considered for the Editor’s Suggestion list.
Reviewer 4 Report
Comments and Suggestions for Authors
Dear Editor,
the present article provides an interesting approach uch to quantify uncertainty regarding compartmental models. The novelty of the introduced method is on an average level, however the overall presentation is promising. You may find attached the review content.

Author Response
Please see the attachement

Round 2
Reviewer 4 Report
Comments and Suggestions for Authors
The manuscript can be accepted for publication.
Regarding comment 8, the authors observed that the uncertainty reaches its maximum when R0 is around 1. This is exactly what I was expected from this Figure. R0 < 1 leads to an asymptotically stable disease-free equilibrium, while R0 > 1 leads to a stable endemic equilibrium. Specifically, the convergence time increases as R0 -> 0 or R0 -> inf, respecively. As a result, when R0 is around 1, a special behavior is encountered where the system oscillates between the 2 equilibria. This phenomenon leads to the maximization of the produced uncertainty.